# Preparation of Chiral Porous Organic Cage Clicked Chiral Stationary Phase for HPLC Enantioseparation

**DOI:** 10.3390/molecules28073235

**Published:** 2023-04-04

**Authors:** Ya-Nan Gong, Qi-Yu Ma, Ying Wang, Jun-Hui Zhang, You-Ping Zhang, Rui-Xue Liang, Bang-Jin Wang, Sheng-Ming Xie, Li-Ming Yuan

**Affiliations:** Department of Chemistry, Yunnan Normal University, Kunming 650500, China

**Keywords:** chiral porous organic cage, high performance liquid chromatography, chiral stationary phase, enantioseparation, thiol-ene click reaction

## Abstract

Porous organic cages (POCs) are a new subclass of porous materials, which are constructed from discrete cage molecules with permanent cavities via weak intermolecular forces. In this study, a novel chiral stationary phase (CSP) has been prepared by chemically binding a [4 + 6]-type chiral POC (C_120_H_96_N_12_O_4_) with thiol-functionalized silica gel using a thiol-ene click reaction and applied to HPLC separations. The column packed with this CSP presented good separation capability for chiral compounds and positional isomers. Thirteen racemates have been enantioseparated on this column, including alcohols, diols, ketones, amines, epoxides, and organic acids. Upon comparison with a previously reported chiral POC NC1-R-based column, commercial Chiralpak AD-H, and Chiralcel OD-H columns, this column is complementary to these three columns in terms of its enantiomeric separation; and can also separate some racemic compounds that cannot be separated by the three columns. In addition, eight positional isomers (iodoaniline, bromoaniline, chloroaniline, dibromobenzene, dichlorobenzene, toluidine, nitrobromobenzene, and nitroaniline) have also been separated. The influences of the injection weight and column temperature on separation have been explored. After the column has undergone multiple injections, the relative standard deviations (RSDs) for the retention time and selectivity were below 1.0 and 1.5%, respectively, indicating the good reproducibility and stability of the column for separation. This work demonstrates that POCs are promising materials for HPLC separation.

## 1. Introduction

Many chemical molecules are chiral. There are two or more enantiomers of one chiral molecule. The resolution of chiral compounds is very important in many fields, particularly in pharmaceutical manufacturing because different enantiomers of a chiral pharmaceutical have different biological, pharmacological, and toxicological effects in organisms [1,2]. However, the resolution of enantiomers is also a challenging task due to their identical non-optical physicochemical properties. Among the many chiral separation methods reported to date, high-performance liquid chromatography (HPLC) using various chiral stationary phases (CSPs) has been proven to be an effective approach for chiral separation due to its advantages of high separation efficiency and wide range of application [3,4,5,6,7]. The enantioselectivity of the CSP is key to this approach. At present, a lot of CSPs have been employed for HPLC enantioseparation, such as polysaccharides [4,6,8], cinchona alkaloids [9,10], chiral crown ethers [11], cyclodextrins [12,13,14], and macrocyclic antibiotics [15,16]. However, most CSPs have some drawbacks, such as limited enantioselectivity and high production cost. Hence, the development of versatile CSPs with exceptional performance has always been a hot topic in chiral separation research.

The use of novel porous materials including metal-organic frameworks (MOFs) [17,18,19], covalent organic frameworks (COFs) [20,21,22,23], and porous organic polymers (POPs) [24,25] as chromatographic stationary phases for separations has attracted a lot of research interest in recent years. These porous materials exhibit good separation performance due to their peculiar characteristics, such as high specific surface areas, defined pore structures, controllable pore sizes, and diverse topological structures. In general, such porous materials are expanded to frameworks or networks via strong covalent bonds or coordination bonds. Therefore, they are insoluble and inconvenient for chemical modification, coating, and immobilization on a chromatographic matrix, which will affect the preparation of efficient chromatographic columns or stationary phases. For instance, when they serve as selectors for the preparation of capillary columns or HPLC stationary phases, their particle suspensions are usually used for coating or bonding onto the inner wall of the column or chromatographic matrix, rather than using homogeneous solutions, which undoubtedly affects the column preparation, column efficiency, reproducibility, separation performance, etc. Porous organic cages (POCs) are a novel subclass of porous materials, which are composed of discrete, cage-like molecules with intrinsic cavities. The discrete molecules assemble into porous solids via weak intermolecular interactions instead of strong covalent bonds or coordination bonds [26,27,28]. When compared to the above-mentioned porous frameworks or networks (MOFs, COFs, and POPs), the most important characteristic of POCs is their good solubility, which enables them to have good solution processability. POCs have widely developed applications, including gas adsorption and storage [29,30,31], catalysis [32,33,34], separation [35,36,37], sensing [38,39,40], and bioimaging [41,42]. The development of POCs as CSPs for chromatographic chiral separation has also received a great deal of attention. In recent years, our group previously reported many chiral POCs as CSPs for gas chromatography (GC) chiral separation with good enantioselectivity [43,44,45,46,47]. More recently, a chiral POC NC1-R used as a selector to prepare CSP for HPLC enantioseparation has also been reported by our group, which demonstrates the potential application of POCs in HPLC [48]. Nevertheless, the utilization of chiral POCs for HPLC enantioseparation is still at a very early stage. Thus, the exploration of more and more chiral POCs with different geometrical structures, pore channels, window sizes, and cavities for the preparation of CSPs for HPLC is of great significance.

Click chemistry is a synthetic concept, which was first proposed by Sharpless in 2001 [49]. The aim of click chemistry is to rapidly and reliably accomplish the chemical synthesis of target molecules via the concatenation of small units. Click chemistry has been extensively used in many fields due to its merits of high reaction efficiency, mild conditions, high selectivity, and low by-products formation. Click reaction has also been widely used to prepare chromatographic stationary phases, such as thiol-ene click reaction and azide-alkyne click reaction [10,11,50]. Herein, we report the use of a [4 + 6]-type chiral POC as a selector to prepare a CSP using a thiol-ene click reaction to separate racemates and positional isomers in HPLC. The chiral POC (C_120_H_96_N_12_O_4_) was synthesized via the Schiff-base condensation of 2-hydroxy-1,3,5-benzenetrialdehyde with (1*R*, 2*R*)-1,2-diphenyl-1,2-ethylenediamine [51], which was clicked onto thiol-functionalized silica gel to prepare the CSP. The CSP packed column showed good selectivity toward various racemates and disubstituted benzene isomers. The influence of the injection mass and column temperature on the separation, repeatability, and stability of the column have been explored.

## 2. Results and Discussion

### 2.1. Characterization of the Prepared Chiral POC and CSP

A variety of methods were used to characterize the chiral POC and CSP, including FT-IR, MALDI-TOF MS, NMR, TGA, and elemental analyses. From the FT-IR spectrum (Figure 1a), we can see the sharp and strong absorption at 1636 cm^−1^ was attributed to the imine bonds (-C=N-) in the chiral POC. The absorption at 3420 cm^−1^ represents the stretching vibration of phenolic hydroxyl (-OH), and the absorption peaks at 3062 and 2866 cm^−1^ can be ascribed to N=C-H and saturated C-H stretching vibrations, respectively. In addition, absorption bands at 3031, 1600, 1490, and 1456 cm^−1^ are the characteristic absorption peaks of the benzene ring. The molecular weight computed from the molecular formula of the chiral POC (C_120_H_96_N_12_O_4_) is 1769.77. MALDI-TOF-MS (Figure 1b) exhibits peaks at *m*/*z* = 1770.70 and 885.86, which correspond to the ions peaks of [M + H]^+^ and [M + 2H]^2+^, respectively. The contents of C, H, and N calculated from the molecular formula (C_120_H_96_N_12_O_4_) are C 81.42%, H 5.47%, and N 9.50%; our elemental analysis found C 81.13%, H 5.36%, and N 9.52%. The NMR data of POC: ^1^H NMR (500 MHz, CDCl_3,_ ppm) δ: 14.51–14.22 (ddd, 4H), 8.90–8.84 (m, 4H), 8.66–8.58 (m, 4H), 8.40–8.37 (m, 4H), 8.30–8.17 (m, 8H), 7.22–7.06 (m, 6H), 4.86–4.66 (m, 12H). All of our characterization data fully support the successful synthesis of the chiral POC.

The fourier transform infrared spectra obtained for the thiol-functionalized silica gel (SiO_2_-SH), CSP and chiral POC are shown in Figure 2a. When compared with SiO_2_-SH, the intensity of the absorption peaks was obviously strengthened at 1636 and 2866 cm^−1^, and new absorption peaks appeared at 1490 and 1456 cm^−1^ in the CSP, indicating that the POC cage molecules were successfully bonded onto the surface of SiO_2_-SH. The TGA curves (Figure 2b) show that, when compared with SiO_2_-SH, the weight of the CSP decreased significantly, further demonstrating that the chiral POC was attached to the surface of SiO_2_-SH. Moreover, the elemental analyses of the C, H, and N content in SiO_2_, SiO_2_-SH, and CSP are shown in Appendix A. The C, H, and N contents in the CSP were obviously higher than those in SiO_2_-SH, which also proved the successful binding of the chiral POC onto SiO_2_-SH. The surface-bound quantity of the chiral POC onto SiO_2_-SH was determined to be 0.18 μmol m^−2^ according to Equation (1) (see Section 3.6).

### 2.2. Resolution of Racemates

To evaluate the chiral separation performance of the CSP, some chiral compounds were separated on the packed column using different proportions of *n*-hexane/isopropanol as the mobile phase. Thirteen racemic compounds were resolved on the column, including alcohols, diols, organic acids, ketones, amines, and epoxides. The chromatographic data obtained for the resolved racemates including the retention factor (*k*_1_), separation factor (*α*), and resolution (*Rs*), were calculated and shown in Table 1. The resulting chromatograms are presented in Figure 3. Table 1 and Figure 3 show that this column exhibits good separation performance for some racemic mixtures, which have high *Rs* values, for instance, the resolution of 1-(1-naphthyl)ethanol (*Rs* = 4.10), 3-benzyloxy-1,2-propanediol (*Rs* = 3.45), 2-phenyl-1-propanol (*Rs* = 2.30), 1-phenylethylamine (*Rs* = 3.72), and ofloxacin (*Rs* = 6.95), demonstrating the good enantioselectivity of the CSP.

The 13 chiral compounds tested were also separated on a previously reported chiral POC NC1-R-based CSP packed column, Chiralpak AD-H, and Chiralcel OD-H columns for comparison (Table 1). There are 4 racemates, namely 2-phenyl-1-propanol, ofloxacin, naproxen, and 1-(3-methylphenyl)ethanol, which cannot be separated on the NC1-R-based CSP packed column (Table 1). In addition, some racemates cannot be separated on the Chiralpak AD-H and Chiralcel OD-H columns, while they can be well separated on this column: 1-(1-naphthyl)ethanol, hydrobenzoin, 1-phenylethylamine, ofloxacin, and naproxen cannot be separated on the Chiralpak AD-H column; hydrobenzoin, ofloxacin, mandelic acid, and naproxen cannot be separated on the Chiralcel OD-H column. Moreover, some racemates can be separated on this column with higher *Rs* and *α* values than on the NC1-R, Chiralpak AD-H, and Chiralcel OD-H columns (Table 1). Some of the chromatograms obtained on these columns are compared in Appendix A. The results indicate that this column can be complementary to the previously reported chiral POC NC1-R, Chiralpak AD-H, and Chiralcel OD-H columns in terms of enantiomeric separation, which can separate some chiral compounds that cannot be separated using the three columns.

The good chiral resolution ability of the CSP was well correlated with the molecular structure of the chiral POC. The synthesized chiral POC molecule has a tetrahedral structure and subtriangular pore windows on each face (Appendix A). The discrete cage molecules are packed into the porous solid via weak intermolecular forces. Due to the enantiomerically pure (1*R*, 2*R*)-1,2-diphenyl-1,2-ethylenediamine serving as one building unit, each molecule of the POC is chiral. Appendix A shows there is an internal cavity in each molecular center. Analytes can access the cavity through the pore window and abundant host–guest interactions will occur. For example, the CSP offers good enantioselectivity for the separation of chiral alcohols, in which hydrogen bonding interactions formed between the hydroxyl groups of the racemates and N, and O atoms of the chiral POC probably play an important role. In addition, other interactions formed between the chiral POC molecule and analytes, such as dipole-dipole, π-π, and van der Waals interactions, are also crucial for chiral separation. Overall, the good enantioselectivity of the chiral POC-based CSP results from its unique molecular structure (i.e., internal cavity and cage-like molecular structure), by which host-guest inclusion, hydrogen bonding, dipole-dipole, π-π, and van der Waals interactions play crucial roles in the chiral separation.

### 2.3. Separation of Positional Isomers

Disubstituted benzene isomers have important applications in industrial production and each positional isomer has unique applications. Therefore, the separation of such positional isomers is particularly important. Nevertheless, positional isomers are difficult to separate due to their analogous physicochemical properties, such as functional groups, boiling point, and dimensions. The CSP also shows a good separation performance toward some disubstituted benzene isomers. Table 2 and Figure 4 show that eight positional isomers (iodoaniline, bromoaniline, chloroaniline, dibromobenzene, dichlorobenzene, toluidine, nitrobromobenzene, and nitroaniline) have been separated on this column. Some positional isomers were separated with higher resolution values, such as the separation of *o*-*/m*-iodoaniline (*Rs* = 3.26), *o*-/*m*-bromoaniline (*Rs* = 3.82), *m*-/*o*-dibromobenzene (*Rs* = 8.03), and *m*-/*o*-dichlorobenzene (*Rs* = 9.66). Our results show that the CSP has a good separation capability toward positional isomers and certain application prospects.

### 2.4. Effect of Injection Mass on Separation

In order to investigate the effect of the injection amount on the separation, various amounts (from 1 to 20 µg) of racemic hydrobenzoin and isomeric iodoaniline were injected into the column. The obtained chromatograms are presented in Figure 5. The results show that the chromatographic peak area increases linearly with an increase in the injection amount (Figure 5), whereas the retention times were almost unchanged. At the same time, the half-peak width increases with an increase in the injection amount and the selectivity decreased (Figure 5). Therefore, an appropriate injection amount should be selected during the separation and analysis of the samples. To achieve better separation effects, the injection mass of hydrobenzoin and iodoaniline should not exceed 10 µg and 5 µg, respectively.

### 2.5. Effect of the Column Temperature on Separation

To evaluate the effect of the column temperature on separation, racemic hydrobenzoin and isomeric iodoaniline were injected into the column and separated using a column temperature in the range of 20–45 °C. It can be seen from Figure 6 that the retention time of hydrobenzoin and iodoaniline on the column decreases as the temperature increases and the resolution value decreases accordingly, which indicates that the separations were exothermic processes.

The Van’t Hoff curves obtained for hydrobenzoin and iodoaniline using this column at different column temperatures are shown in Figure 7. A good linear correlation between ln*k*’ and 1/T demonstrates that the separation mechanism did not change at the column temperatures studied. The Gibbs free energy change (Δ*G*), enthalpy change (Δ*H*), and entropy change (Δ*S*) of the analytes were calculated and shown in Table 3 (see Section 3.7 for the computation formula used for Δ*G*, Δ*H*, and Δ*S*). Table 3 shows the ΔG values of *R*,*R*-/*S*,*S*-hydrobenzoin and *o*, *m*, *p*-iodoaniline were negative, indicating that the transfer of the samples from the mobile phase to the CSP was a thermodynamically spontaneous process. The more negative Δ*G* of the analyte, the easier it transfers from the mobile phase to the CSP, which will result in a longer retention time. Table 3 shows Δ*G*_(*R*,*R*-hydrobenzoin)_ < Δ*G*_(*S*,*S*-hydrobenzoin)_ and Δ*G*_(*p*-iodoaniline)_ < Δ*G*_(*m*-iodoaniline)_ < Δ*G*_(*o*-iodoaniline)_. Theoretically, the retention time sequence was *R*, *R*-hydrobenzoin > *S*, *S*-hydrobenzoin and *p*-iodoaniline > *m*-iodoaniline > *o*-iodoaniline, which was in accordance with the chromatograms (Figure 6).

### 2.6. Reproducibility and Stability of the Column

The excellent stability and reproducibility of a chromatographic column are important for its practical application. To study the reproducibility and stability of the column, the separation of hydrobenzoin and iodoaniline was performed after different times of injection. Appendix A shows the chromatograms obtained for hydrobenzoin and iodoaniline on the column after it has undergone 10, 100, 200, and 300 injections. Appendix A shows there are no significant changes in the retention times and selectivities, and the relative standard deviations (RSDs) were <1.0% and <1.5% for the retention time and selectivity, respectively, demonstrating that this column has good repeatability and stability for separation.

## 3. Materials and Methods

### 3.1. Chemicals and Reagents

Phenol, (1*R*, 2*R*)-1,2-diphenyl-1,2-ethanediamine, hexamethylenetetramine (HMTA), 1-allylimidazole, 1,4-dibromobutane, azodiisobutyronitrile (AIBN), (3-mercaptopropyl) trimethoxysilane, trifluoroacetic acid (TFA) and K_2_CO_3_ were obtained from Adamas-beta (Shanghai, China). Commercial spherical silica (5 µm, 300 m^2^ g^−1^) was provided from Suzhou Nanwei Technology Co., Ltd. (Suzhou, China). Other commonly used solvents, such as N, N-dimethylformamide (DMF), chloroform, toluene, acetonitrile, ethanol, and pyridine, were obtained from Tianjin Fengchuan Chemical Reagent Technology Co., Ltd. (Tianjin, China). Racemic 1-(3-methylphenyl) ethanol, 1-(1-naphthyl) ethanol, 1-(4-chlorophenyl) ethanol, mandelic acid, naproxen and ofloxacin were purchased from Adamas-beta (Shanghai, China); 2-phenyl-1-propanol, 3-benzyloxy-1,2-propanediol, hydrobenzoin, 1-phenyl-1-propanol, 1-phenylethylamine, benzoin, and *trans*-1,2-diphenylethylene oxide were obtained from Aladdin (Shanghai, China). Positional isomers, such as iodoaniline, bromoaniline, chloroaniline, dichlorobenzene, dibromobenzene, toluidine, nitroaniline, and nitrobromobenzene were provided from Adamas-beta (Shanghai, China).

### 3.2. Instrumentation

A Shimadzu LC-16 (Shimazu, Kyoto, Japan) HPLC equipped with an SPD-16 UV-Vis detector and LC-16 pump was used in this work. A HPLC stainless-steel column (250 mm × 2.1 mm i.d.) and Alltech slurry packer (Alltech, Deerfield, IL, USA) were used for column packing. A Bruker DRX 500 spectrometer (Bruker, Bremen, Germany) was used to obtain the nuclear magnetic resonance (NMR) spectra. An SDT-650 thermal analyzer (TA Instruments, New Castle, DE, USA) was used to obtain the thermogravimetric analysis data. Mass spectrometry was carried out on a Bruker UltrafleXtreme MALDI-TOF mass spectrometer (Thermo Fisher, Bremen, Germany). Elemental analysis data were obtained on a Vario EL III elemental analyzer (Elementar, Langenselbold, Germany).

### 3.3. Synthesis of Chiral POC

The chiral POC was prepared according to the approach reported by M. Petryk et al. (Figure 8) [51]. The 2-hydroxy-1,3,5-benzenetrialdehyde building block was first synthesized (see Appendix A). (1*R*, 2*R*)-1,2-Diphenyl-1,2-ethanediamine (0.64 g, 3 mmol) was dissolved in 160 mL of H_2_O/DMSO (90/10, *v*/*v*) upon stirring. After 5 min, 2-hydroxy-1,3,5-benzenetrialdehyde (0.36 g, 2 mmol) and TFA (15 µL) were added, and the resulting mixture was stirred at room temperature for 14 days. A yellow solid was generated during the reaction process. The mixture was filtered and the solid washed three times with chloroform and H_2_O, and then dried at 70 °C under vacuum for 6 h. Finally, the solid was recrystallized via slow diffusion of diethyl ether into a solution of the crude product in chloroform to afford the pure POC crystals (0.30 g, yield 35%).

### 3.4. Preparation of the Chiral POC-Based CSP

Preparation of the CSP was carried out following the procedure described in Figure 9. The chiral POC was reacted with 1,4-dibromobutane and 1-allylimidazole to synthesize the alkenyl-functionalized chiral POC. The alkenyl-functionalized products were then attached to the thiol-functionalized silica gel using a thiol-ene click reaction. The detailed preparation process of the CSP was as follows:

#### 3.4.1. Preparation of Thiol-Functionalized Silica Gel

Commercial spherical silica (5.0 g) was added to a 10% HCl solution, and the resulting mixture was stirred at 100 °C for 24 h. The resulting mixture was filtered and washed to neutral with deionized water, and then dried under vacuum at 180 °C to afford activated silica gel.

Activated silica (2.5 g) was dispersed in 50 mL of anhydrous toluene, and (3-mercaptopropyl)trimethoxysilane (2.0 mL) and anhydrous pyridine (0.5 mL) were then successively added. The resulting mixture was heated at reflux under the protection of an N_2_ atmosphere. After 48 h, the reaction mixture was filtered and the resulting solid was washed with toluene, methanol, and acetone in sequence. The solid was then dried at 100 °C under vacuum for 6 h to obtain thiol-functionalized silica gel.

#### 3.4.2. Synthesis of Alkenyl-Functionalized Chiral POC

Typically, a 250 mL round-bottom flask was charged with the synthesized chiral POC (0.5 g, 0.28 mmol) and potassium carbonate (0.2 g, 1.7 mmol), and 40 mL of acetonitrile was then added. After stirring for 5 min, 1,4-dibromobutane (0.2 mL) was added and the resulting mixture was heated at reflux for 48 h. The mixture was filtered, and the filtrate was concentrated in vacuo on a rotary evaporator to obtain a brown solid. The crude product was repeatedly washed with *n*-hexane and ultrapure water, and then dried at 80 °C under vacuum to obtain compound **1** (0.56 g).

Compound **1** (0.54 g) was dispersed in acetonitrile (60 mL) with the aid of ultrasound for 10 min, and 1-allylimidazole (0.3 mL) was then added. Subsequently, the reaction mixture was heated at 60 °C under a N_2_ atmosphere for 48 h. The resulting solution was concentrated to ~20 mL and then poured into *n*-hexane (20 mL) to form a precipitate. The solids were collected by filtration and washed with *n*-hexane, ultrapure water, and acetone, and then dried at 60 °C under vacuum to yield the alkenyl-functionalized chiral POC (compound **2**) (0.58 g).

#### 3.4.3. Bonding of Alkenyl-Functionalized Chiral POC onto Thiol-Functionalized Silica Gel

The alkenyl-functionalized chiral POC (compound **2**, 0.54 g) and methanol (40 mL) were charged into a 100 mL round-bottom flask with stirring. After 30 min, thiol-functionalized silica gel (1.3 g) and AIBN (50 mg) were added, and then heated at 65 °C under a N_2_ atmosphere. After reaction for 48 h, the resulting mixture was filtered, and the solid was washed with methanol and acetone. Finally, the solid was dried at 80 °C under vacuum to give the CSP.

### 3.5. Column Packing

The prepared CSP was packed into a HPLC stainless steel column using a slurry-packing technique. Firstly, the CSP (1.2 g) was uniformly dispersed in 20 mL of *n*-hexane/isopropanol (90/10, *v*/*v*). Then, using *n*-hexane/isopropanol (90/10, *v*/*v*) as the driving solvent, the suspensions was packed into the HPLC column using a slurry packer at a pressure of 30 MPa for 30 min.

### 3.6. Calculation of the Surface-Bound Amount of the CSP

The surface-bound amount of the chiral POC on the CSP was calculated according to the following formula (Equation (1)) [12,52]:(1)$$\frac{\mu mol}{m^{2}}=\frac{C\%\times 10^{6}}{S\times 12.001\times N_{c}\times (100-\frac{C\%}{N_{c}\times 12.001}\times M_{r})}$$
where *C*% represents the increased carbon content compared to the thiol-functionalized silica gel, *N_c_* represents the number of carbon atoms in the bonding portion, *M_r_* represents the molecular weight of the bonding portion, and S represents the surface area of silica gel (300 m^2^ g^−1^).

### 3.7. Calculation of the Thermodynamic Parameters

The enthalpy change (Δ*H*), entropy change (Δ*S*), and Gibbs free energy change (Δ*G*) were calculated according to the following equations (Equations (2) and (3)) [53]:(2)$$\ln\;k\prime =-\frac{\Delta H}{RT}+\frac{\Delta S}{R}+\ln\;\Phi$$
(3)ΔG=ΔH−TΔS
where *k*′ is the retention factor, *R* is the gas constant, *T* is the absolute temperature, and *Φ* is the phase ratio. *k*′ was calculated according to (Equation (4)).
(4)$$k\prime =\frac{t_{R}-t_{0}}{t_{0}}$$
where *t_R_* is the retention time of the enantiomers and *t_0_* is the dead time.

## 4. Conclusions

We have prepared a POC-based chiral CSP for HPLC separations. The experimental results demonstrate that the column allows to achieve a good separation of some racemic compounds and positional isomers. Thirteen racemic compounds belonging to various classes and eight positional isomers were separated on the CSP packed column. The column can be complementary to the previously reported chiral POC NC1-R-based column, Chiralpak AD-H and Chiralcel OD-H columns in terms of their enantioseparation, and can separate some racemates that cannot be separated using the three previously reported chromatographic columns. In addition, the column exhibited good stability and repeatability for separation after multiple injections. This work indicates that POC has great potential in HPLC separation.

## Figures and Tables

**Figure 1 molecules-28-03235-f001:**
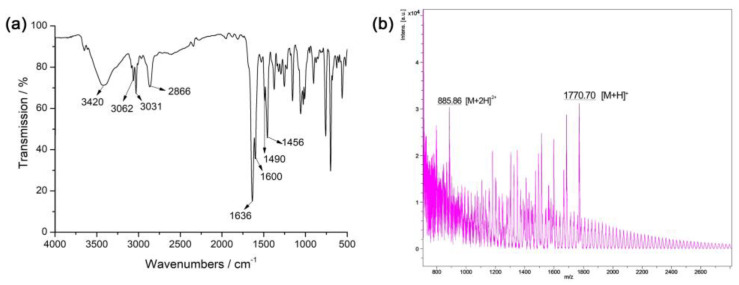
(**a**) FT–IR and (**b**) MALDI–TOF–MS spectra obtained for the chiral POC.

**Figure 2 molecules-28-03235-f002:**
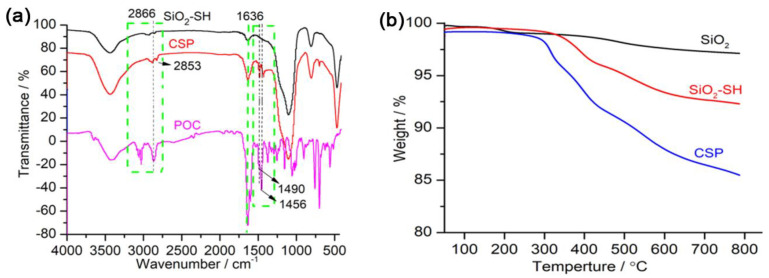
(**a**) FT–IR spectra obtained for the prepared chiral POC, SiO_2_–SH, and CSP. (**b**) TGA curves of the SiO_2_, SiO_2_–SH, and CSP.

**Figure 3 molecules-28-03235-f003:**
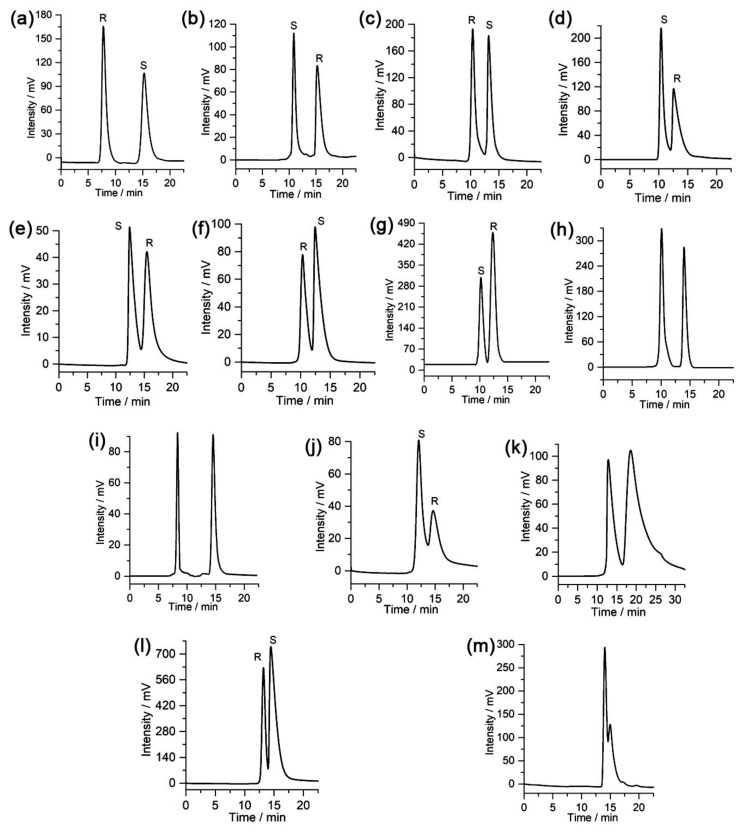
Chromatograms obtained for the separation of racemates on the chiral POC-based CSP packed column: (**a**) 1-(1-Naphthyl)ethanol, (**b**) 3-benzyloxy-1,2-propanediol, (**c**) 2-phenyl-1-propanol, (**d**) *trans*-1,2-diphenylethylene oxide, (**e**) benzoin, (**f**) 1-phenyl-1-propanol, (**g**) hydrobenzoin, (**h**) 1-phenylethylamine, (**i**) ofloxacin, (**j**) mandelic acid, (**k**) naproxen, (**l**) 1-(4-chlorophenyl)ethanol, and (**m**) 1-(3-methylphenyl)ethanol.

**Figure 4 molecules-28-03235-f004:**
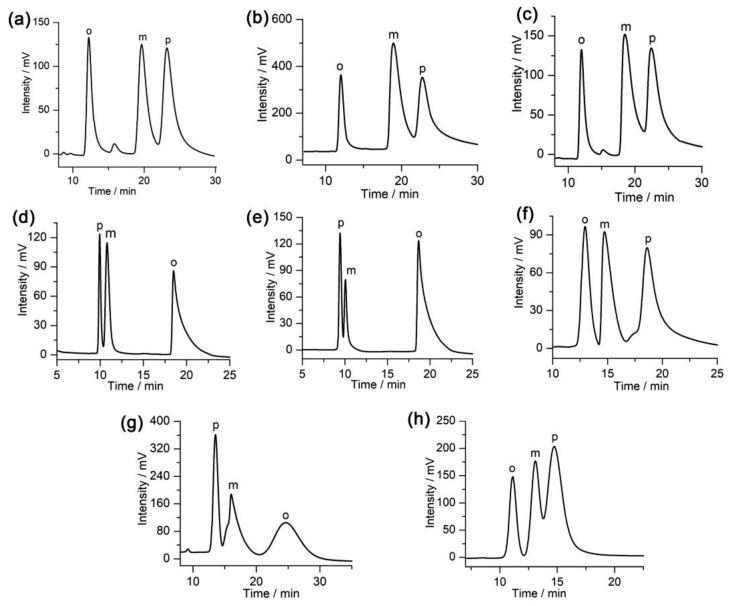
Chromatograms obtained for the separation of positional isomers on the chiral POC-based CSP packed column: (**a**) iodoaniline, (**b**) bromoaniline, (**c**) chloroaniline, (**d**) dibromobenzene, (**e**) dichlorobenzene, (**f**) toluidine, (**g**) nitrobromobenzene, and (**h**) nitroaniline.

**Figure 5 molecules-28-03235-f005:**
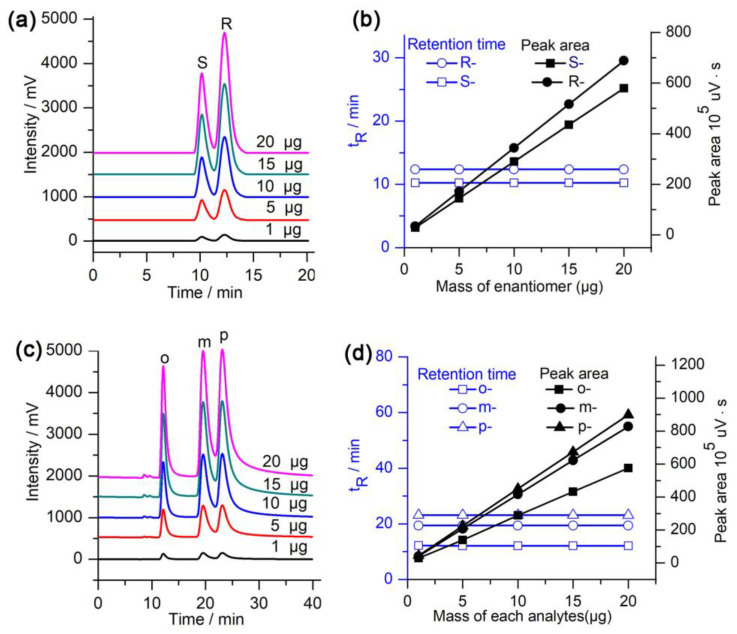
Effect of the injection amount on the separation: (**a**,**c**) Chromatograms obtained for with various amounts of hydrobenzoin and iodoaniline and (**b**,**d**) effect of various amounts of hydrobenzoin and iodoaniline on the retention time and peak area. Other chromatographic conditions are the same as those in Table 1 and Table 2.

**Figure 6 molecules-28-03235-f006:**
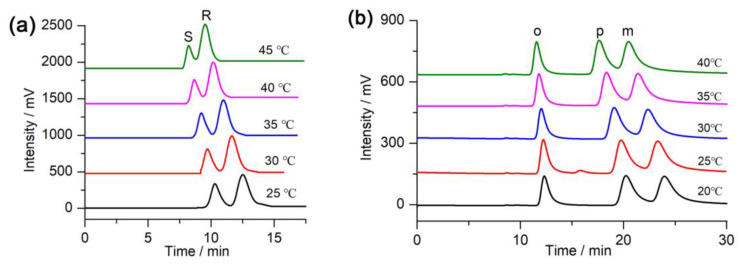
Separation of (**a**) hydrobenzoin and (**b**) iodoaniline on the column at different column temperatures. The other chromatographic conditions are the same as those in Table 1 and Table 2.

**Figure 7 molecules-28-03235-f007:**
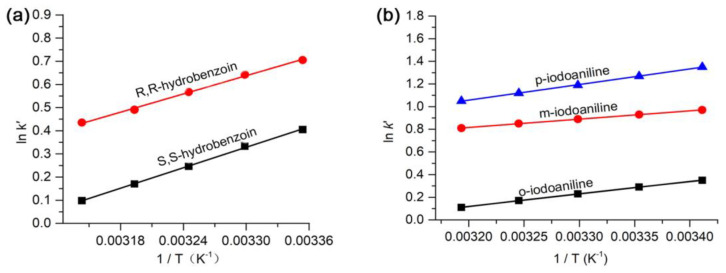
Van’t Hoff curves obtained for (**a**) hydrobenzoin and (**b**) iodoaniline.

**Figure 8 molecules-28-03235-f008:**
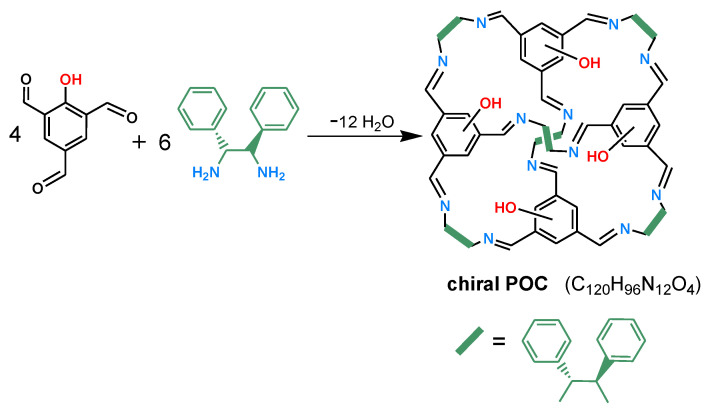
Synthesis of the chiral POC.

**Figure 9 molecules-28-03235-f009:**
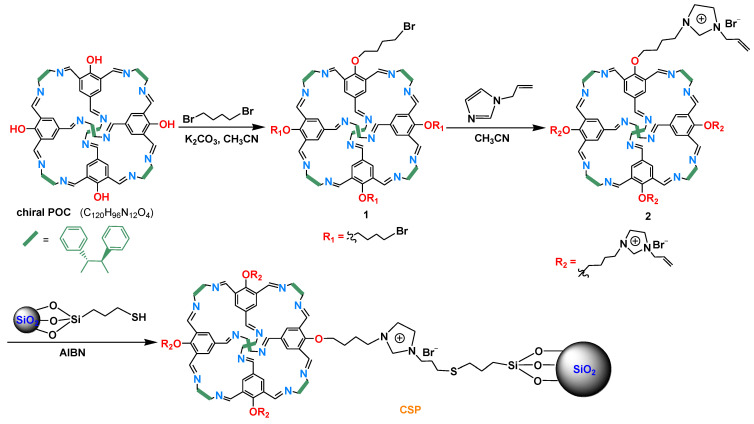
Preparation of the CSP.

**Table 1 molecules-28-03235-t001:** Resolution of racemates on the chiral POC NC1-R-based column, Chiralpak AD-H, and Chiralcel OD-H columns.

Racemates	This Column			NC1-R Column [48]			Chiralpak AD-H Column			Chiralcel OD-H Column		
	*k* _1_	*α*	*Rs*	*k* _1_	*α*	*Rs*	*k* _1_	*α*	*Rs*	*k* _1_	*α*	*Rs*
1-(1-Naphthyl)ethanol ^a^	1.11	4.80	4.10	1.28	2.09	4.09	1.47	1.00	^_ d^	1.93	1.14	1.23
3-Benzyloxy-1,2-propanediol ^a^	1.63	1.87	3.45	4.65	1.63	5.47	2.24	1.19	0.70	2.73	1.06	2.12
2-Phenyl-1-propanol ^b^	1.27	1.62	2.30	1.10	1.00	^_ d^	1.36	1.00	^_ d^	1.52	1.16	1.82
*trans*-1,2-Diphenylethylene oxide ^a^	1.17	1.47	1.38	0.54	1.33	0.74	0.86	2.76	19.8	0.83	1.95	4.38
Benzoin ^a^	1.67	1.46	1.31	3.41	1.08	1.26	4.70	1.36	6.96	2.52	1.58	6.53
1-Phenyl-1-propanol ^a^	1.14	1.47	1.38	1.04	1.50	2.04	0.83	1.06	0.63	0.83	1.11	0.50
Hydrobenzoin ^a^	1.22	1.62	1.81	0.76	1.57	2.50	0.36	1.00	^_ d^	2.62	1.00	^_ d^
1-Phenylethylamine ^c^	1.41	1.90	3.72	0.86	1.37	1.88	0.71	1.00	^_ d^	0.96	1.19	0.81
Ofloxacin ^d^	1.01	3.45	6.95	0.92	1.00	^_ d^	0.55	1.00	^_ d^	0.90	1.00	^_ d^
Mandelic acid ^a^	1.52	1.41	1.02	0.77	1.73	3.13	3.36	1.20	4.55	2.20	1.06	0.62
Naproxen ^a^	2.20	1.81	1.33	2.01	1.00	^_ d^	3.10	1.00	^_ d^	1.07	1.00	^_ d^
1-(4-Chlorophenyl)ethanol ^a^	1.49	1.17	1.12	1.13	1.51	3.07	0.86	1.04	0.49	0.78	1.13	0.67
1-(3-Methylphenyl)ethanol ^a^	1.58	1.11	0.34	1.08	1.00	^_ d^	0.93	1.00	^_ d^	0.75	1.25	1.26

Experimental conditions: column temperature, 25 °C; flow rate, 0.1 mL min^−1^. ^a^ Mobile phase: *n*-hexane/isopropanol = 90/10 (*v*/*v*); ^b^ mobile phase: *n*-hexane/isopropanol = 85/15 (*v*/*v*); ^c^ mobile phase: *n*-hexane/isopropanol = 95/5 (*v*/*v*); *n*-hexane/isopropanol = 99/1 (*v*/*v*); ^d^: could not be separated.

**Table 2 molecules-28-03235-t002:** Separation of positional isomers on the chiral POC column.

Positional Isomers	Separation Factor (*α*)		Resolution (*Rs*)	
	*α* _1_	*α* _2_	*Rs* _1_	*Rs* _2_
Iodoaniline ^a^	2.17	1.25	3.26	1.52
Bromoaniline ^a^	2.11	1.29	3.82	1.45
Chloroaniline ^a^	2.05	1.31	2.95	1.31
Dibromobenzene ^a^	1.21	2.54	1.42	8.03
Dichlorobenzene ^a^	1.17	3.02	1.31	9.66
Toluidine ^a^	1.25	1.44	1.60	1.77
Nitrobromobenzene ^b^	1.32	1.84	1.58	1.90
Nitroaniline ^a^	1.37	1.84	1.74	0.89

Experimental conditions: column temperature, 25 °C; flow rate, 0.1 mL min^−1^. ^a^ Mobile phase: *n*-hexane/isopropanol = 90/10 (*v*/*v*); ^b^ mobile phase: *n*-hexane/isopropanol = 99/1 (*v*/*v*).

**Table 3 molecules-28-03235-t003:** Δ*H*, Δ*S*, Δ*G* and correlation coefficient (*R*^2^) values for the separation of hydrobenzoin and iodoaniline.

Analytes	Δ*H* (kJ mol^−1^)	Δ*S* (J mol^−1^ K^−1^)	Δ*G* (kJ mol^−1^)	*R* ^2^
*S*,*S*-Hydrobenzoin	−10.91 ± 0.31	−31.62 ± 0.99	−1.48 ± 0.31	0.9987
*R*,*R*-Hydrobenzoin	−12.25 ± 0.22	−35.60 ± 0.72	−1.64 ± 0.22	0.9968
*o*-Iodoaniline	−6.41 ± 0.10	−20.18 ± 0.34	−3.53 ± 0.10	0.9986
*m*-Iodoaniline	−9.14 ± 0.07	−4.16 ± 0.22	−4.87 ± 0.07	0.9986
*p*-Iodoaniline	−11.23 ± 0.11	−5.64 ± 0.38	−10.79 ± 0.11	0.9974

## Data Availability

All relevant data are included in the article.

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
