# Peer review of "Preparation of Chiral Porous Organic Cage Clicked Chiral Stationary Phase for HPLC Enantioseparation"

_molecules, 2023, doi:10.3390/molecules28073235_

Round 1

Reviewer 1 Report

This MS reports about the preparation of a novel chiral stationary phase (CSP) made of a chiral porous organic cage that was packed into a column and tested for the separation of a series of chiral compounds (including also some  pharmaceuticals). Although the subject of the MS is interesting for a broad range of readers, I think that an extensive revision of english style and language is required prior publication.

Some suggestions:

- l. 10: define the acronym CSP

- l. 12 and elsewhere: please do not say "CSP packed column", but rather column packed with the CSP

- l. 15: you have not defined POC-NC1-R yet

- l. 36: applications

- l. 92: please avoid the use of first person

- l. 127: including

- ll. 128-133: please revise this sentence since it is not well written

- ll. 145-149: please revise this sentence and check english style

- ll. 158-159: too many repetitions

- ll. 162-165: please revise these sentences

- l. 168: in which

- l. 172: maybe remove "which"?

- l. 200: remove "separation"

- ll. 204-205: this is somehow obvious...

- l. 213: do you mean that measurements have been made in this range of temperatures?

- maybe the experimental section can be moved before results

- l. 300: allows to achieve a good separation of

- l. 30: belonging

- ll. 302-303: can be complementary to?

Author Response

Response to Reviewer 1 Comments

This MS reports about the preparation of a novel chiral stationary phase (CSP) made of a chiral porous organic cage that was packed into a column and tested for the separation of a series of chiral compounds (including also some pharmaceuticals). Although the subject of the MS is interesting for a broad range of readers, I think that an extensive revision of english style and language is required prior publication.

Response: Thank you for your suggestion. The use of English throughout our manuscript has been revised by a professional language editing service (International Science Editing, Bay K, Unit 11a, Shannon Freezone - West, Shannon, Co. Clare, Ireland). We believe that the revision is now acceptable for the review process.

Some suggestions:

- l. 10: define the acronym CSP

Response: CSP is the abbreviation of “chiral stationary phase”. We have defined it in the manuscript.

- l. 12 and elsewhere: please do not say "CSP packed column", but rather column packed with the CSP

Response: Thank you for your suggestion. We have corrected it in the manuscript.

- l. 15: you have not defined POC-NC1-R yet.

Response: POC is the abbreviation of “porous organic cage”. NC1-R is one of the chiral POC, which was synthesized according to Huang et al. (Angew. Chem. Int. Ed. 2017, 56, 7787-7791).

- l. 36: applications

Response: We have corrected it in the manuscript.

- l. 92: please avoid the use of first person

Response: We have corrected it in the manuscript.

- l. 127: including

Response: We have corrected it in the manuscript.

- ll. 128-133: please revise this sentence since it is not well written

Response: This sentence has been revised.

- ll. 145-149: please revise this sentence and check english style

Response: This sentence has been revised.

- ll. 158-159: too many repetitions

Response: This sentence has been revised.

- ll. 162-165: please revise these sentences

Response: This sentence has been revised.

- l. 168: in which

Response: We have corrected it in the manuscript.

- l. 172: maybe remove "which"?

Response: The word “which” has been removed.

- l. 200: remove "separation"

Response: The word “separation” has been removed.

- ll. 204-205: this is somehow obvious...

Response: We have corrected it in the manuscript.

- l. 213: do you mean that measurements have been made in this range of temperatures?

Response: Yes, measurements have been made in this range of temperatures (25-45 ℃).

- maybe the experimental section can be moved before results

Response: Thank you for your suggestion. However, the format required for Molecules is in this order: Introduction, Results and Discussion, Materials and Methods, and Conclusions.

- l. 300: allows to achieve a good separation of

Response: Thank you for your suggestion. We have corrected it in the manuscript.

- l. 30: belonging

Response: We have corrected it in the manuscript.

- ll. 302-303: can be complementary to?

Response: We have corrected it in the manuscript.

Reviewer 2 Report

Zhang, Wang and co-workers report a novel chiral stationary phase (CSP) based on a chiral organic cage and its evaluation for the analytic separation of enantiomers and positional isomers. The CSP was found very effective with a large set of compounds and with comparable performance to the known ones. The CSP was correctly characterized by different techniques. The reported results should be of high interest to the community of analytical and organic chemists. I suggest acceptance in Molecules after minor corrections noted below:

1) The part concerning the synthesis of the CSP (parts 3.3 and 3.4) should be placed before the part regarding its characterization after the last paragraph of the introduction (lines 74 to 87). The whole discussion on synthesis (lines 74-87, §3.3 and §3.4) could be included in the Results and Discussion part.

2) Please avoid words such as “triumphantly” (line 111) and “prosperous” (line 116), they don’t sound very scientific

3) Whereas the English is very good throughout the paper, several mistakes and English grammar errors were noted in part 2.2 (see lines 128-133, 145-159 and 163-172). Please revise this part

4) Lines 204-205, it is written that the appropriate injection mass should be selected. Please indicate what are these values for the two examples of figure 5.

5) line 229: “will have longer …”

6) line 276: TFA

7) Line 281: please indicate the obtained mass of the chiral POC and the yield

8) Please explain why using the 1,3-diazolium ring in the linker between the POC and the silica particle. Why not just using an alkyl chain? Moreover, this linker introduces a charge. Is it necessary?

9) Conclusion: line 301 “belonging”; line 305: remove “also” because “besides” is enough

Author Response

Zhang, Wang and co-workers report a novel chiral stationary phase (CSP) based on a chiral organic cage and its evaluation for the analytic separation of enantiomers and positional isomers. The CSP was found very effective with a large set of compounds and with comparable performance to the known ones. The CSP was correctly characterized by different techniques. The reported results should be of high interest to the community of analytical and organic chemists. I suggest acceptance in Molecules after minor corrections noted below:

1) The part concerning the synthesis of the CSP (parts 3.3 and 3.4) should be placed before the part regarding its characterization after the last paragraph of the introduction (lines 74 to 87). The whole discussion on synthesis (lines 74-87, §3.3 and §3.4) could be included in the Results and Discussion part.

Response: Thank you for your suggestion. However, the format required for Molecules is in this order: Introduction, Results and Discussion, Materials and Methods, and Conclusions.

2) Please avoid words such as “triumphantly” (line 111) and “prosperous” (line 116), they don’t sound very scientific

Response: Thank you for your suggestion. The words “triumphantly” and “prosperous” have been changed to“successfully” and “successful” accordingly.

3) Whereas the English is very good throughout the paper, several mistakes and English grammar errors were noted in part 2.2 (see lines 128-133, 145-159 and 163-172). Please revise this part

Response: We have revised these parts. The use of English throughout our manuscript has been revised by a professional language editing service (International Science Editing, Bay K, Unit 11a, Shannon Freezone - West, Shannon, Co. Clare, Ireland). We believe that the revision is now acceptable for the review process.

4) Lines 204-205, it is written that the appropriate injection mass should be selected. Please indicate what are these values for the two examples of figure 5.

Response: The appropriate injection mass for the two examples of figure 5 have been provided in the manuscript (line 210-211).

5) line 229: “will have longer …”

Response: We have corrected it in the manuscript.

6) line 276: TFA

Response: We have corrected it in the manuscript.

7) Line 281: please indicate the obtained mass of the chiral POC and the yield

Response: The yield has been provided in the manuscript (line 290).

8) Please explain why using the 1,3-diazolium ring in the linker between the POC and the silica particle. Why not just using an alkyl chain? Moreover, this linker introduces a charge. Is it necessary?

Response: Our previously report has proved that the cationic 1,3-diazolium ring plays important role in chiral separation and can enhance the enantioselectivity of the CSP (Y. Wang, et. al. Anal. Chem. 2022, 94, 4961-4969). Besides, many cyclodextrin based CSPs with cationic 1,3-diazolium ring on the spacer can enhance enantioselectivity have also been reported (e.g., X. Yao, et al. J. Chromatogr. A 2014, 1326, 80-88; X. Yao, et al. Anal. Chem. 2016, 88, 4955-4964; X. Li, et al. J. Chromatogr. A 2016, 1467, 279-287; J. Zhou, et al. TrAC-Trend Anal. Chem. 2015, 65, 22-29). The bonding arm have significant impact on chiral separation. The cationic 1,3-diazolium ring can provide some additional interactions during chiral separation, such as electrostatic interaction, which are beneficial for chiral separation.

9) Conclusion: line 301 “belonging”; line 305: remove “also” because “besides” is enough

Response: Thank you for your suggestion. We have corrected it in the manuscript.
